# Neural Harmonics: Bridging Spectral Embedding and Matrix Completion in Self-Supervised Learning

Marina Munkhoeva[†‡]                    Ivan Oseledets [§]

## Abstract

Self-supervised methods received tremendous attention thanks to their seemingly heuristic approach to learning representations that respect the semantics of the data without any apparent supervision in the form of labels. A growing body of literature is already being published in an attempt to build a coherent and theoretically grounded understanding of the workings of a zoo of losses used in modern self-supervised representation learning methods. In this paper, we attempt to provide an understanding from the perspective of a Laplace operator and connect the inductive bias stemming from the augmentation process to a low-rank matrix completion problem. To this end, we leverage the results from low-rank matrix completion to provide theoretical analysis on the convergence of modern SSL methods and a key property that affects their downstream performance.

## 1 Introduction

Self-supervised methods have garnered significant interest due to their heuristic approach to learning representations that capture the semantic information of data without requiring explicit supervision in the form of labels. Contrastive learning based methods among the former would use the repulsion among arbitrary pair of points in the batch, while non-contrastive would rely on the consistency among different views of the same image. While self-supervised representation learning becomes more ubiquitous in the wild, especially in the important domain such as medical imaging [16], the theoretical grounding of these methods would potentially help avoid the pitfalls in applications. Unsurprisingly, researchers are already trying to build theoretically sound understanding of modern self-supervised representation learning methods.

The overall goal of this work is to understand self-supervised representation learning through the lens of nonlinear dimensionality reduction methods (e.g. Laplacian Eigenmaps [3]) and low-rank matrix completion problem [28]. To this end, we take on a Laplace operator perspective on learning the optimal representations in the manifold assumption. We then derive a trace maximization formulation to learn eigenfunctions of the Laplace operator of the underlying data manifold. We adopt the heat kernel based embedding map that in theory under certain conditions is an almost isometric embedding of the low-dimensional manifold into the Euclidean space. As a result, we discuss how existing several SSL methods (e.g. SIMCLR [9], BARLOWTWINS [1], VICREG [2]) can be comprehended under this view.

It is important to note that our current understanding of the topic lacks one crucial aspect. Traditional spectral methods commonly operate with complete kernel matrices, whereas our approach deals with incomplete and potentially noisy ones. The only available similarity information among examples is derived from the data augmentation process, which generates a positive pair. Meanwhile the

---

[†]Correspondence to `marina.munkhoeva@tuebingen.mpg.de`

[‡]Max Planck Institute for Intelligent Systems, Tübingen, Germany

[§]Artificial Intelligence Research Institute (AIRI), Skolkovo Institute of Science and Technology (Skoltech), Moscow, Russian Federation

37th Conference on Neural Information Processing Systems (NeurIPS 2023).

remaining examples in the batch are either seen as negative ones (contrastive) or are not considered at all (non-contrastive).

A pertinent and often overlooked question emerges: how can self-supervised learning methods effectively leverage such limited signals to converge towards meaningful representations? In response, we shed light on this matter by establishing a connection between SSL and a matrix completion problem. We demonstrate that these optimization problems are Lagrangian dual of each other, implying that optimizing the SSL objective simultaneously entails reconstructing the kernel matrix. We can summarize the contributions of this paper as follows:

- We propose an eigen-problem objective for spectral embeddings from graphs induced by augmentations and use it to interpret modern SSL methods.

- We show that SSL methods do *Laplacian-based nonlinear dimensionality reduction* and *low-rank matrix completion* simultaneously. We leverage theory behind matrix completion problem to provide insights on the success of SSL methods and their use in practice.

- While the number of observed entries required by theory decreases with epochs, we find that the actual number is a constant and the former eventually intersects the latter.

- We find a possible explanation for disparity in downstream performance of the backbone and projection outputs.

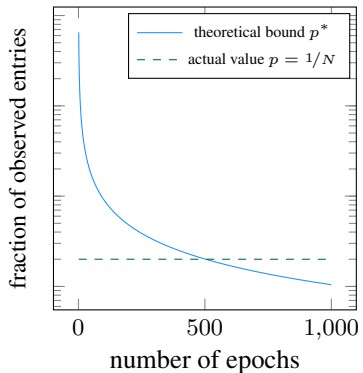

Figure 1: Illustrative graph of the theoretical fraction $p^*$ of the observed entries required for matrix completion to succeed with high probability. As training proceeds, unmaterialized kernel matrix size $n$ increases and $p^*$, which is roughly $\sim \log n/n$, decreases. Eventually, the actual (constant) fraction $p$ of observed entries under self-supervised learning augmentation protocol intersects the theoretical bound.

## 2 Background

This work relies on the manifold hypothesis, a hypothesis that many naturally occurring high-dimensional data lie along a low-dimensional latent manifold inside the high-dimensional space. Since we can only observe a sample from the manifold in the ambient space, neither the true manifold nor its metric are available to us. Although we never explicitly work with Laplace-Beltrami operator, we still give a brief definition below to provide some grounding for the reasoning later.

**Laplace operator** Let $\mathcal{M}$ be a Riemannian manifold and $g$ be the Riemannian metric on $\mathcal{M}$. For any smooth function $u$ on $\mathcal{M}$, the gradient $\nabla u$ is a vector field on $\mathcal{M}$. Let $\nu$ be the Riemannian volume on $\mathcal{M}$, $d\nu = \sqrt{\det g}\, dx^1...dx^D$. By the divergence theorem, for any smooth functions $u$ and $v$ (plus smooth and compact support assumptions) $\int_{\mathcal{M}} u \operatorname{div}\nabla v \, d\nu = -\int_{\mathcal{M}} \langle \nabla u, \nabla v \rangle d\nu$, where $\langle \cdot, \cdot \rangle = g(\cdot, \cdot)$. The operator $\Delta = \operatorname{div}\nabla$ is called the Laplace-Beltrami operator of the Riemannian manifold $\mathcal{M}$.

In practice, we usually work with finite samples, and Laplace operator is typically approximated with graphs or meshes. While the latter are typically used in computational mathematics, the former find a widespread use in machine learning. Below is a brief overview of the relevant graph notions, for a detailed exposition please see [36].

**Graph Laplacian and Spectral Embedding** Given a graph $\Gamma = (V, E)$ with $|V| = n$ vertices and a set of edges $e_{ij} = (v_i, v_j)$ that form a weighted adjacency matrix $\mathbf{A}_{ij} = w_{ij}$ with non-negative weight $w_{ij} \geq 0$, whenever there is $e_{ij} \in E$, otherwise 0. With a degree matrix $\mathbf{D} = \operatorname{diag}(\mathbf{A1})$, the graph Laplacian is given by $\mathbf{L} = \mathbf{D} - \mathbf{A}$, the corresponding random walk Laplacian is a normalization $\mathbf{L}_{rw} = \mathbf{I} - \mathbf{D}^{-1}\mathbf{A}$. All graph Laplacians admit an eigenvalue decomposition, i.e. $\mathbf{L} = \mathbf{U}\Lambda\mathbf{U}^{\top}$, where $\mathbf{U} \in \mathbb{R}^{n \times n}$ contains eigenvectors in columns and a diagonal matrix $\Lambda \in \mathbb{R}^{n \times n}$ has eigenvalues on the diagonal. Note that there is a trivial eigenpair $(0, \mathbf{1})$. [5, 4] show that the eigenvectors of the graph Laplacian of a point-cloud dataset converge to the eigenfunctions of the Laplace-Beltrami operator under uniform sampling assumption.

Whenever one has an affinity matrix, a positive semidefinite pairwise similarity relations among points, the classical way to obtain embeddings for the points is to perform spectral embedding. A typical algorithm would include constructing a graph based on the affinity matrix, computing $k$ first eigenvectors of its Laplacian and setting the embeddings to be the rows of the matrix $\mathbf{U} \in \mathbb{R}^{n \times k}$ that contains the eigenvectors as columns.

**Matrix completion**  Let $\mathbf{M} \in \mathbb{R}^{n \times n}$ is partially observed matrix with rank $r$. Under Bernoulli model, each entry $\mathbf{M}_{ij}$ is observed independently of all others with probability $p$. Let $\Omega$ be the set of observed indices. The matrix completion problem aims to recover $\mathbf{M}$ from $m = |\Omega|$ observations. The standard way to solve this problem is via nuclear norm minimization:

$$\min_{\mathbf{X} \in \mathbb{R}^{n \times n}} \|\mathbf{X}\|_* \quad \text{subject to} \quad \mathbf{X}_{ij} = \mathbf{M}_{ij} \text{ for } (i,j) \in \Omega, \tag{1}$$

where $\|\mathbf{X}\|_*$ is the nuclear norm of $\mathbf{X}$, i.e. the sum of its singular values. A large body of work [8, 28, 27, 10] has succeeded in providing and enhancing of the conditions that guarantee the optimal solution $\mathbf{X}^*$ to be both unique and equal to $\mathbf{M}$ with high probability.

**Notation**  Any matrix $\mathbf{X} \in \mathbb{R}^{n_1 \times n_2}$ has a singular value decomposition (SVD) $\mathbf{X} = \mathbf{U}\Sigma\mathbf{V}^\top$, where the columns of the matrix $\mathbf{U} \in \mathbb{R}^{n_1 \times n_1}$ are left singular vectors, the singular values $\sigma_1 \geq \sigma_2 \geq \cdots \geq \sigma_{\min(n_1, n_2)}$ lie on the diagonal of a diagonal matrix $\Sigma \in \mathbb{R}^{n_1 \times n_2}$, and right singular vectors are the columns of $\mathbf{V} \in \mathbb{R}^{n_2 \times n_2}$. $\|\mathbf{X}\|_F$, $\|\mathbf{X}\| = \sigma_1$, $\|\mathbf{X}\|_* = \sum_i \sigma_i$ denote Frobenius, spectral and nuclear norms of $\mathbf{X}$, respectively. $\|\mathbf{x}\|_p$ denotes $p$-th vector norm.

## 3  SSL and Spectral Embedding

In this section, we will provide a construction that covers most of the self-supervised learning methods and gives SSL a novel interpretation. First, we formalize the setup from the perspective of the manifold hypothesis. Then, we make some modelling choices to yield an SSL formulation as a trace maximization problem, a form of eigenvalue problem. Finally, we describe how three well known representatives SSL methods fall under this formulation.

Let $\mathbf{X} \in \mathbb{R}^{n \times d'}$ be a set of $n$ points in $\mathbb{R}^{d'}$ sampled from a low-dimensional data manifold observed in a $d'$-dimensional ambient Euclidean space. However, data is rarely given in the space where each dimension is meaningful, in other words $d' \gg d^*$, where $d^*$ is unknown true dimensionality of the manifold. The goal of nonlinear dimensionality reduction is to find a useful embedding map into $d$-dimensional Euclidean space with $d \ll d'$. Two of the classical approaches, namely Eigenmaps [3] and Diffusion maps [11], use eigenfunctions of the graph Laplacian, an approximation of the Laplace operator associated with the data manifold. Both can be loosely described as a variant of map

$$\Phi(\mathbf{x}) = [\, \phi_1(\mathbf{x}) \quad \phi_2(\mathbf{x}) \quad \ldots \quad \phi_d(\mathbf{x}) \,], \tag{2}$$

where $\phi_k$ is $k$-th eigenfunction of the negative Laplace operator on the underlying manifold.

This type of map bears a direct connection to theoretical question how to find an embedding of certain types of manifolds into Euclidean spaces, studied in differential geometry. For instance, [6] construct embedding for a given smooth manifold by using its heat kernel. This approach has been subsequently enhanced through the application of a truncated heat kernel expansion. Furthering this trajectory, [26] explore whether and when such embedding is close to being isometric, which is desirable as isometric embedding preserves distances.

Motivated by the heat kernel embedding literature, we provide a general construction for self-supervised methods in what follows. The heat kernel on a manifold can be represented as an expansion $H(p, q, t) = \sum_{k=0}^{\infty} e^{-\lambda_k t} \phi_k(p)\phi_k(q)$, where $p$ and $q$ are some points on the manifold, $t > 0$ and $\phi_k$ are normalized eigenfunctions of the Laplace operator, i.e. $-\Delta\phi_k = \lambda_k\phi_k$ and $|\phi_k|_2 = 1$. However, working with finite data, we can only operate with a graph approximation of the manifold, and will make use of the heat kernel construction for graphs. The latter is given by a matrix exponential $\mathbf{H}_t = e^{-t\mathbf{L}}$, alternatively represented through an expansion $\mathbf{H}_t = \sum_{i=0}^{n} e^{-\lambda_k t} \phi_k \phi_k^T$, where $\mathbf{L}$ is a graph Laplacian and $(\lambda_k, \phi_k)$ is $k$-th eigenpair of $\mathbf{L}$.

Consider Laplacian EigenMaps method, it first constructs a graph and weighs the edges with a heat kernel of the Euclidean space which takes the form of a Gaussian kernel

$h_E(\mathbf{x}, \mathbf{y}, t) = \exp(-||\mathbf{x} - \mathbf{y}||_2^2/t)$ whenever the distance is less then some hyperparameter, i.e. $||\mathbf{x} - \mathbf{y}||_2 < \epsilon$, or $\mathbf{y}$ is among $k$ nearest neighbours of $\mathbf{x}$. Next, the method proceeds with finding the eigenvectors of the graph Laplacian constructed from the thus weighted adjacency matrix $\mathbf{W}$.

Contrarily, the proposed construction acknowledges the cardinality and complexity typically associated with the modern day datasets, where it is challenging to select the cutoff distance or the number of neighbours, and impractical to compute all pairwise distances. Our ideal graph has edges reflecting the semantic similarity between points; e.g. there is an edge whenever the pair of points belong to the same class. The drawback is that this matrix is unknown. This is where augmentation comes into play as it provides a peek into only a fraction of the entries in the unmaterialized heat kernel matrix. Consequently, we claim that SSL implicitly solves a low-rank matrix completion problem by instantiating some of the pairwise similarity entries in $\mathbf{H}_t$ via the augmentation process. However, prior to this, we need to demonstrate the generality of the perspective we have formulated. To accomplish this, we articulate a trace maximization problem.

## 3.1 Trace Minimization

To start, we construct an unmaterialized heat kernel matrix in accordance with the standard self-supervised learning protocol for augmentations. Given a dataset with $N$ points, one training epoch generates $a$ views of each point. Typically, $a = 2$ and training runs for $n_{epochs}$ number of epochs. The whole training generates exactly $n_{epochs} \times a$ views per original instance. As a result the total number of processed examples is $n = N \times a \times n_{epochs}$, thereby we have that $\mathbf{H}_t \in \mathbb{R}^{n \times n}$.

As a rule, SSL utilises architectures with a heavy inductive bias, e.g. ResNet [19] / Vision Transformers [14] for image data. Thus, we may safely assume that the same instance views are close in the embedding space and are connected. This results in a block diagonal adjacency matrix $\mathbf{W}$, where $i$-th block of ones accounts for all the views of $i$-th image among $N$ original images. The fraction of observed entries equal to $p = N \times (n_{epochs} \times a)^2/n^2 = 1/N$, a constant fraction for a given dataset.

Note that in the ideal scenario, we would know the cluster/class affiliation for each point and would have connected same-class views with edges. However, in reality, we only know instance affiliation of each view. Let us denote the ideal scenario heat kernel matrix as $\mathbf{H}_t$ and its partially observed counterpart used in reality — $\widehat{\mathbf{H}}_t$.

To obtain the heat kernel matrix, we use the normalized random walk Laplacian $\mathbf{L}_{rw}$, for which the heat kernel matrix is as before: $\mathbf{H}_t = \exp(-t\mathbf{L}_{rw}) = \sum_{k=0}^{n} \exp(-\lambda_k t)\mathbf{u}_k\mathbf{u}_k^\top$, where $\mathbf{u}_k$ is $k$-th eigenvector and $\lambda_k$ a corresponding eigenvalue of $\mathbf{L}_{rw}$. To obtain spectral embeddings from an ideal $\mathbf{H}_t$, one would then proceed with solving a trace minimization form of eigenvalue problem in (3), where the optimal solution is exactly the eigenvectors of $\mathbf{L}_{rw}$, i.e. $\mathbf{Z}^* = \mathbf{U}$.

$$\max_{\mathbf{Z}} \quad \mathrm{Tr}(\mathbf{Z}^\top \mathbf{H}_t \mathbf{Z}) \qquad\qquad (3) \qquad\qquad \max_{\theta} \quad \mathrm{Tr}(\mathbf{Z}_\theta^\top \widehat{\mathbf{H}}_t \mathbf{Z}_\theta) \qquad\qquad (4)$$
$$\text{s.t.} \quad \mathbf{Z}^\top \mathbf{Z} = \mathbf{I}, \qquad\qquad\qquad\qquad \text{s.t.} \quad \mathbf{Z}_\theta^\top \mathbf{Z}_\theta = \mathbf{I}_d, \ \mathbf{Z}_\theta^\top \mathbf{1} = \mathbf{0},$$

However, for lack of a better alternative we resort to the incomplete $\widehat{\mathbf{H}}_t$ and need to learn a parameterized map $\mathcal{F}_\theta(\mathbf{X}) = \mathbf{Z}_\theta$ in (4). Apart from the specified eigenfunction pairwise orthogonality constraint, the resulting loss is comprised of implicit constraints on the trivial eigenfunction (identity) and function norm. But before we treat (3) with incomplete $\mathbf{H}_t$ as matrix completion problem, we show that this formulation can be seen as a generalization for a number of existing self-supervised learning methods. Specifically, several modelling choices differentiate the resultant methods.

## 3.2 Self-Supervised Methods Learn Eigenfunctions of the Laplace Operator

**SIMCLR** Consider the following contrastive loss, variants of which are widely adopted in self-supervised learning methods: $L_{CL} = \sum_{i,j \in +pairs} l_{ij} + l_{ji}$, where

$$l_{ij} = \log \frac{e^{\kappa\langle \mathbf{z}_i, \mathbf{z}_j \rangle}}{\sum_{k \neq i} e^{\kappa\langle \mathbf{z}_i, \mathbf{z}_k \rangle}} = \kappa\langle \mathbf{z}_i, \mathbf{z}_j \rangle - \log \sum_{k \neq i} e^{\kappa\langle \mathbf{z}_i, \mathbf{z}_k \rangle}, \qquad\qquad (5)$$

with $\mathbf{z}_i = f_\theta(\mathbf{x}_i)$ is an embedding of input $\mathbf{x}_i$ given by function $f$ parameterized by a neural network with learnable parameters $\theta$, which produces unit-norm embeddings, i.e. $\|\mathbf{z}_i\|_2 = 1$. One iteration of SIMCLR takes a batch of $N$ data points and creates $2N$ views, augmenting each sample to obtain a positive pair $(\mathbf{z}_i, \mathbf{z}_j)$, while treating all the other samples and their augmentations as contrastive samples $\mathbf{z}_k$.

Let us include $k = i$ in the sum in denominator of (5) for a moment. A goal of SIMCLR is to obtain optimal representations $\mathbf{Z} \in \mathbb{R}^{2N \times d}$ such that for each positive pair $(i, j)$, their representations will be aligned as much as possible $\langle \mathbf{z}_i, \mathbf{z}_j \rangle \to 1$. Let $\mathbf{A}_{ij} = \exp(\kappa[\mathbf{Z}\mathbf{Z}^\top]_{ij})$ and $\mathbf{D} = \mathrm{diag}(\mathbf{A}\mathbf{1})$, then we can rewrite (5) as

$$l_{ij} = \log \mathbf{A}_{ij} - \log \sum_{k=1}^{2N} \mathbf{A}_{ik} = \log \mathbf{A}_{ij} - \log \mathbf{D}_{ii} = \log \frac{\mathbf{A}_{ij}}{\mathbf{D}_{ii}} = \log[\mathbf{D}^{-1}\mathbf{A}]_{ij} \tag{6}$$

where we are interested in the right hand side $\log[\mathbf{D}^{-1}\mathbf{A}]_{ij}$. Let us note that $[\mathbf{D}^{-1}\mathbf{A}]_{ij}$ from (6) is a normalized adjacency of a graph $G = (V, E)$, where node set $V$ is the set of views. Since representations live on a unit $(d-1)$-sphere, we have a choice of naturally arising distributions (von Mises-Fisher, spherical normal, etc) to instantiate the weighting function $\mu(\mathbf{x}_i, \mathbf{x}_j)$. The model choice of SIMCLR is to instantiate the weighting function with the density of the von Mises-Fisher distribution with $\kappa > 0$ (without the partition function):

$$\mathbf{A}_{ij} = \exp(\kappa \mathbf{z}_i^\top \mathbf{z}_j), \tag{7}$$

where $\mathbf{A}_{ij}$ equals $e^\kappa$, with a typical value $\kappa = 2$, whenever $i, j$ is a positive pair, and $e^0 = 1$ otherwise. Note that $\log(\mathbf{D}^{-1}\mathbf{A}) = \mathbf{D}^{-1}\mathbf{A} - \mathbf{I} + o((\mathbf{D}^{-1}\mathbf{A})^2) \approx -\mathbf{L}_{rw}$, thus in loose terms the objective in (5) maybe be seen as a minimization of a trace of a negative of the graph Laplacian by learning $\mathbf{z}$'s that shape $\mathbf{A}$.

Decoupled Contrastive Learning [38] can be seen as reweighting the adjacency matrix by setting $\mathbf{A}_{ij} = w(z_i, z_j)$, where $w(z_i, z_j)$ is a reweighting function to emphasize hard examples.

For non-contrastive methods (BARLOWTWINS, VICREG, etc) SSL methods (where no negative examples are used, e.g. BARLOWTWINS, SIMSIAM), the respective losses could also be interpreted as learning a diffusion map. The key distinction with contrastive methods expresses itself in setting $A_{ij} = 0$ for view pairs $(i, j)$ that do not have shared original, thus eliminating any signal from a possibly negative pair.

**BARLOWTWINS** method normalizes the columns of representation matrices $\mathbf{Z}_a, \mathbf{Z}_b \in \mathbb{R}^{N \times d}$, which is the same as the function norm constraint in (4). The overall objective is as follows:

$$J = \sum_{ii} (\mathbf{Z}_a^\top \mathbf{Z}_b - \mathbf{I})_{ii}^2 + \alpha \sum_{i \neq j} (\mathbf{Z}_a^\top \mathbf{Z}_b - \mathbf{I})_{ij}^2, \tag{8}$$

and simultaneously aims at three things: (i) enforce the closeness of the rows, i.e. positive pair embeddings, (ii) restrict the norm of the columns, i.e. the eigenfunctions, (iii) orthogonalize columns, again the eigenfunctions. The kernel matrix choice in BARLOWTWINS is a simple bi-diagonal adjacency matrix $\mathbf{A}_{ij} = 1$ as long as $(i, j)$ is a positive pair, and 0 otherwise.

**VICREG** objective is a weighted sum of *variance, covariance* and *invariance* terms:

$$J_{var} = \sum_{k=1}^{d} \max(0, 1 - \sqrt{[\mathbf{Z}^\top \mathbf{Z}]_{kk}}), \quad J_{cov} = \sum_{k \neq l} [\mathbf{Z}^\top \mathbf{Z}]_{kl}^2, \quad J_{inv} = \sum_{ij} \mathbf{A}_{ij} \|\mathbf{Z}_i - \mathbf{Z}_j\|^2,$$

a similar formulation to the previous method, however, the choice for the adjacency matrix here is little bit different. Individual terms of this loss have separate coefficients. The choice of these hyperparameters defines the implicit adjacency matrix entries, controls the maximum function norm allowed and the trade-off between orthogonality and the trace terms in (4).

## 4   SSL and Low-Rank Matrix Completion

In this section, we use low-rank matrix completion theory to show that the limited information from augmentations might be quite enough under certain conditions. First, we establish the connection between our self-supervised learning objective (4) and the low-rank matrix completion problem. We then draw practical insights from the existing convergence theory for matrix completion problem.

## 4.1 Low-Rank Matrix Completion Dual

We argue that the objective in (3) with a substitute incomplete kernel matrix $\widehat{\mathbf{H}}_t$ implicitly contains an objective for the low-rank matrix completion problem. To show this, we first introduce the general form of the nuclear norm minimization problem with affine subspace constraint in (9).

$$\begin{aligned}\min_{\mathbf{X}} \quad & \|\mathbf{X}\|_* \\ \text{subject to} \quad & \mathcal{A}(\mathbf{X}) = \mathbf{b},\end{aligned} \quad (9) \qquad\qquad \begin{aligned}\max_{\mathbf{q}} \quad & \mathbf{b}^\top \mathbf{q} \\ \text{subject to} \quad & \|\mathcal{A}^*(\mathbf{q})\| \leq 1,\end{aligned} \quad (10)$$

The subspace is given by linear equations $\mathcal{A}(\mathbf{X}) = \mathbf{b}$ and linear operator $\mathcal{A} : \mathbb{R}^{n \times n} \to \mathbb{R}^p$ can be a random sampling operator. The dual for the nuclear norm $|\cdot|_*$ is the operator norm $|\cdot|$. The problem in (9) has been initially introduced as a heuristic method for seeking minimum rank solutions to linear matrix equations, but has later been shown to have theoretical guarantees under certain conditions on the linear operator $\mathcal{A}$ [28, 27, 32, 17]. The Lagrangian dual of (9) is given by (10), where operator $\mathcal{A}^* : \mathbb{R}^p \to \mathbb{R}^{n \times n}$ is the adjoint of $\mathcal{A}$.

We write down an instance of (9) in (11) below to better reflect the specifics of our setting. Since the true underlying heat kernel matrix $\mathbf{H}_t$ is observed only *partially* with known entries indicated by a collection of index pairs $\Omega$ induced by augmentation process, we can form a sampling symmetric matrix $\mathbf{W}$: $\mathbf{W}_{ij} = 1$ if $(i, j) \in \Omega$, and 0 otherwise, indicating observed entries. Now, the incomplete kernel matrix can be written down explicitly as $\widehat{\mathbf{H}} = \mathbf{W} \odot \mathbf{H}$, where $\odot$ denotes Hadamard (element-wise) product. The constraint in (9) instantiates as $\mathcal{A}(\mathbf{X}) = \texttt{vec}(\mathbf{W} \odot \mathbf{X})$.

$$\begin{aligned}\min_{\mathbf{X} \in \mathcal{S}_+} \quad & \|\mathbf{X}\|_* \\ \text{subject to} \quad & \texttt{vec}(\mathbf{W} \odot \mathbf{X}) = \texttt{vec}(\widehat{\mathbf{H}})\end{aligned} \quad (11) \qquad \begin{aligned}\max_{\mathbf{Z} \in \mathbb{R}^{n \times d}} \quad & \operatorname{Tr} \mathbf{Z}^\top \widehat{\mathbf{H}} \mathbf{Z} \\ \text{subject to} \quad & \mathbf{Z}^\top \mathbf{Z} = \mathbf{I}\end{aligned} \quad (12)$$

We proceed by showing that maximisation of the trace formulation in (12) embraces reconstruction of the incomplete underlying kernel matrix with entries known to us only from the augmentation process and specified by the matrix $\mathbf{W}$.

**Proposition 4.1.** *The trace maximization problem given in* (12) *is a Lagrangian dual of low-rank matrix completion problem in* (11).

*Proof.* First, we show that (12) is an instance of (10). Let linear operator $\mathcal{A}(\mathbf{X}) : \mathbb{R}^{n \times n} \to \mathbb{R}^{n^2}$ be a sampling vectorization of $\texttt{vec}(\mathbf{W} \odot \mathbf{X})$. By trace of a matrix product, we can rewrite the objective as $\operatorname{Tr}[\mathbf{Z}^\top \widehat{\mathbf{H}} \mathbf{Z}] = \operatorname{Tr}[\widehat{\mathbf{H}} \mathbf{Z} \mathbf{Z}^\top] = (\texttt{vec}\widehat{\mathbf{H}})^\top \texttt{vec}(\mathbf{Z} \mathbf{Z}^\top) = \mathbf{b}^\top \mathbf{q}$.

The adjoint operator $\mathcal{A}^*(\mathbf{q}) = \texttt{mat}(\mathbf{D}_{\mathbf{W}} \mathbf{q})$ acts a sampling matricization, i.e. it samples and maps $\mathbf{q}$ back to $\mathbb{R}^{n \times n}$: $\mathcal{A}^*(\mathbf{q}) = \mathbf{W} \odot \mathbf{Z} \mathbf{Z}^\top$, and the constraint of the dual (10) becomes $\|\mathbf{W} \odot \mathbf{Z} \mathbf{Z}^\top\| \leq 1$. As $\mathbf{Z}$ admits singular value decomposition $\mathbf{Z} = \mathbf{U} \Sigma \mathbf{V}^\top$, the constraint in (12) $\mathbf{Z}^\top \mathbf{Z} = \mathbf{I}$ implies $\sigma_i(\mathbf{Z}) = 1$ for all $i$.

To establish equivalence of the constraints in (10) and (12), i.e. $\|\mathbf{W} \odot \mathbf{Z} \mathbf{Z}^\top\| = \sigma_1(\mathbf{W} \odot \mathbf{Z} \mathbf{Z}^\top) \leq 1$ given $\mathbf{W}$ and $\sigma_1(\mathbf{Z}) = 1$, we can use a result for singular values of Hadamard product [], specifically, for $k = 1, 2, \ldots, n$: $\sum_{i=1}^k \sigma_i(\mathbf{A} \odot \mathbf{B}) \leq \sum_{i=1}^k \min(c_i(\mathbf{A}), r_i(\mathbf{A})) \sigma_i(\mathbf{B})$, where $c_i(\mathbf{A})$ and $r_i(\mathbf{A})$ are column and row lengths of $\mathbf{A}$, and $\sigma_1 \geq \sigma_2 \geq \cdots \geq \sigma_n$. Let $\mathbf{A} = \mathbf{W}$ and $\mathbf{B} = \mathbf{Z} \mathbf{Z}^\top$, then $r_i = c_i = \sqrt{K}$, yielding $\sigma_1(\mathbf{W} \odot \mathbf{Z} \mathbf{Z}^\top) \leq \sqrt{K}$, where $K = n_{epochs} \times a$ is a constant, consequently, it can be accounted for by rescaling $\mathbf{W}$ and not affecting the maximization. Finally, since (11) is an instance of (9), we can conclude that (12) is dual to (11). $\square$

This result allows us to make use of theoretical guarantees obtained in matrix completion in what follows.

## 4.2 Convergence

**Incoherence** Standard incoherence [7, 23, 20, 17, 27] is the key notion in the analysis of matrix completion problem. Intuitively, incoherence characterises the ability to extract information from a small random subsample of columns in the matrix. More formally, it is defined as an extent to which the singular vectors are aligned with the standard basis.

**Definition 4.2.** Given matrix $\mathbf{M} \in \mathbb{R}^{n_1 \times n_2}$ with rank $r$ and SVD $\mathbf{M} = \mathbf{U\Sigma V}^\top$, $\mathbf{M}$ is said to satisfy the *standard incoherence* condition with coherence parameter $\mu$ if

$$\max_{1 \leq i \leq n_1} \|\mathbf{U}^\top \mathbf{e}_i\|_2 \leq \sqrt{\frac{\mu r}{n_1}} , \quad \max_{1 \leq j \leq n_2} \|\mathbf{V}^\top \mathbf{e}_j\|_2 \leq \sqrt{\frac{\mu r}{n_2}} , \tag{13}$$

where $\mathbf{e}_i$ is the $i$-th standard basis vector of a respective dimension.

Since the matrix we aim to recover is symmetric, $n_1 = n_2 = n$ and its left, right singular and eigenvectors are identical. The coherence parameter $\mu = \frac{n}{r} \max_i \|\mathbf{Ue}_i\|_2^2$ range from 1 (incoherent) to $\frac{n}{r}$ (coherent).

To the best of our knowledge, optimal sample complexity bounds for matrix recovery via nuclear norm minimisation were obtained in [10]. Specifically, given $\mathbf{M}$ satisfies standard incoherence condition (13) with parameter $\mu$, if uniform sampling probability $p^* \geq c_0 \mu r \log^2(n)/n$ for some $c_0 > 0$, then $\mathbf{M}$ is the unique solution to (1) with high probability.

Matrix recovery is easy when it is highly incoherent — when information is spread more uniformly among its columns/rows, loosing a random subset of its entries is not as big of a deal as when information is concentrated in certain important columns/rows. On the other hand, high incoherence intuitively makes matrices harder when used as feature matrices in downstream tasks. This might explain why typical SSL methods rely on the output of backbone network (*representations*) rather than the output of projection head (*embeddings*).

It is easy to see that downstream performance depends on the alignment of the target matrix $\mathbf{Y}$ with the left eigenvectors of the feature matrix $\mathbf{X}$. The target matrix $\mathbf{Y}$ in downstream classification task is typically a very simple binary matrix, and can be shown to have low incoherence. However, whenever $\mathbf{X}$ is obtained as the projection head output of a network learned via self-supervised learning method with a spectral embedding type objective, the incoherence of $\mathbf{X}$ is inherently tied to the incoherence of the kernel matrix. The latter needs to have high incoherence to be recoverable. Consequently, we put forward the following proposition and find its empirical support in Section 5.2.

**Proposition 4.3.** *Projection head outputs (embeddings) yield lower performance on the downstream task due to their high incoherence. The complexity of the projection head correlates with coherence of the backbone output (representations).*

**Only a fraction of total entries in $\mathbf{H}_t$ is required for matrix recovery with high probability.** We can further narrow down the bound on $p^*$ if we consider structured matrix completion problem in the form of some side information which has a direct connection to self-supervised setup. Let the column/row space of $\mathbf{M}$ lie in some known $r'$-dimensional subspace of $\mathbb{R}^n$ spanned by the columns of $\bar{\mathbf{U}}$, $n > r' \geq r$. Then the nuclear norm minimization problem transforms into:

$$\min_{\mathbf{X}} \quad \|\mathbf{X}\|_* \quad \text{subject to} \quad (\bar{\mathbf{U}}\mathbf{X}\bar{\mathbf{U}}^\top)_{ij} = \mathbf{M}_{ij}, \quad (i,j) \in \Omega. \tag{14}$$

In practice, we use neural network parameterisation to learn the heat kernel map, this choice inadvertently restricts the reconstructed kernel to be aligned with the column space of the network outputs, bringing the inductive bias of the architecture into picture.

The optimal sampling complexity bound for (14) extends as $p^* \gtrsim \mu \bar{\mu} r \bar{r} \log(\bar{\mu}\bar{r}) \log n / n^2$, where $\bar{\mu}$ and $\bar{r}$ are the coherence and the rank of $\bar{\mathbf{U}}$, respectively. Suppose we wanted to recover some binary adjacency matrix $\mathbf{A}$, such that $\mathbf{A}_{ij} = 1$ if $i, j$ belong to the same cluster, 0 otherwise. Because $\mathbf{A}$ can be rearranged to be block-diagonal with $r$ blocks, its coherence $\mu(\mathbf{A}) = n/rn_{min}$, and exact reconstruction is possible provided

$$p^* \gtrsim \bar{\mu}\bar{r} \log(\bar{\mu}\bar{r}) \log n / nn_{min}, \tag{15}$$

where $n_{min}$ is the minimal cluster size. Heat kernel matrix constructed from such $\mathbf{A}$ will have its eigenspectrum closely resembling that of $\mathbf{A}$, albeit smooth, yet still having same pattern in eigengaps. So we may safely adopt this bound for $\mathbf{H}_t$. For balanced class datasets $n_{min} = n/c$, and we can immediately see that the number of required observations $m = p^* n^2 = c\bar{\mu}\bar{r} \log(\bar{\mu}\bar{r}) \log n$ grows linearly with the number of classes $c$ in the dataset.

For illustrative purposes we plot the theoretical bound $p^*$ on the fraction of the observed entries for a successful matrix completion from (15) in Figure 1 along with the actual fraction $p$ of observed

Table 1: Performance comparison for the trace maximization formulation (RQMIN) and VICREG. Mean and standard deviation for validation set accuracy across 5-10 runs for CIFAR-10, CIFAR-100 and ImageNet-100.

| | CIFAR-10 | | CIFAR-100 | | ImageNet-100 | |
|---|---|---|---|---|---|---|
| | top-1 | top-5 | top-1 | top-5 | top-1 | top-5 |
| VICREG | $91.15_{\pm 0.16}$ | $99.64_{\pm 0.05}$ | $67.57_{\pm 0.20}$ | $89.90_{\pm 0.13}$ | $78.89_{\pm 0.38}$ | $93.94_{\pm 0.17}$ |
| RQMIN | $91.19_{\pm 0.13}$ | $99.67_{\pm 0.04}$ | $68.12_{\pm 0.26}$ | $90.12_{\pm 0.11}$ | $78.98_{\pm 0.33}$ | $94.45_{\pm 0.23}$ |

entries under self-supervised learning augmentation protocol to demonstrate that the latter intercepts the former given enough training epochs. To be specific, we set the size of the training dataset $N = 50k$ (CIFAR-10 size), the cluster size (number of same class examples) $n_{min} = 5000$ ($c = 10$), the number of views $a = 2$, the number of epochs $n_{epochs}$ range from 1 to 1000, and $r = 512$ (embedding size), assume $\mu = 20$ (which seems to be a fair estimate in light of the experiments in Section 5) and $c_0 = 5$, a constant used to control the probability of exact matrix recovery.

Based on this bound, we highlight the following factors that play important role in the success of any SSL method with spectral embedding type objective. The choice of the similarity function affects the incoherence parameter in the bound. The number of samples per class (alternatively the minimal cluster size $n_{min}$) should also be high enough for $p^*$ to decrease rapidly. Finally, though potentially in contrast to the empirical observations (higher $d$ on ImageNet yields higher downstream performance), the rank $r$, effectively the dimension of embedding $d$, should not be too large.

# 5 Experiments

First, we verify that the performance of the proposed formulation in (4) and the corresponding loss function, denoted RQMIN, is at least on par with the state-of-the-art methods. We then study the effect of the complexity of the projection head on the incoherence and its connection with the downstream performance of the backbone against projection head outputs.

Here we report the training hyperparameters for all of the experiments. As VICReg is extremely sensitive to the choice of hyperparameters (e.g. increasing learning rate with increased batch size negatively affects training – learning diverges), we adopt the same hyperparameters for training RQMIN for a fair comparison. We follow the standard VICReg protocol adopted and finetuned for CIFAR-10/100 and ImageNet-100 in the library for self-supervised learning methods for visual representation learning *solo-learn* [12].

We train ResNet-18 backbone architecture with 3-layer MLP projection head (respective hidden dimensions: 2048-2048-2048). The batch size is 256 for CIFAR datasets and 512 for ImageNet-100. For pretraining, the learning rate schedule is linear warm-up for 10 epochs and cosine annealing, the optimizer is LARS with learning rate 0.3. For linear probe training, SGD with step learning rate schedule with steps at 60 and 80 epochs. The number of pre-training epochs is 1000 for CIFAR and 400 for ImageNet-100, downstream training – 100 epochs.

## 5.1 Comparable performance

To demonstrate that our trace maximization objective is on par with existing SSL methods, we test our objective in (4) on a standard ResNet-18 backbone neural network with 3-layer MLP projection head and obtain comparable results on CIFAR-10, CIFAR-100, and ImageNet-100 to state-of-art methods. As most of the latter yield almost identical performance given enough hyperparameter tuning, we pick VICREG as a representative to compare against. Following the standard protocol with 1000 and 400 pre-training epochs for both CIFAR datasets and for ImageNet-100, respectively, linear probe for downstream evaluation is trained for 100 epochs. We do not tune hyperparameters and use default values. Mean and standard deviation for downstream accuracy across 5-10 trials with different seed values are reported in Table 1.

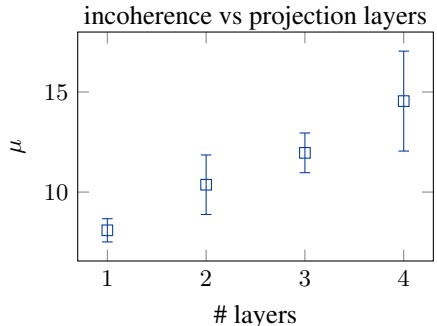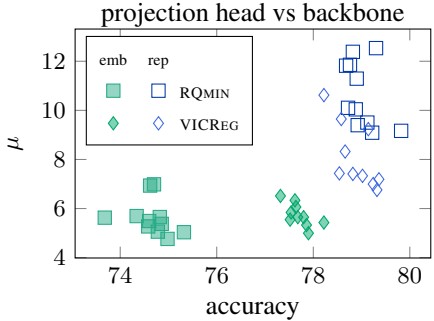

Figure 2: **(left)** Coherence of the backbone outputs (*representations*) grows with the increasing number of layers in the projection head. **(right)** Downstream accuracy versus incoherence. Green marks show downstream performance and incoherence of embeddings, blue marks – representations. Embeddings and representations of the same type of model are almost always separable in these coordinates: embeddings exhibit lower performance and lower coherence (small $\mu$) while representations perform better and have higher coherence (large $\mu$).

## 5.2 Incoherence effect on downstream performance

To test whether incoherence could explain the performance disparity between the outputs of the backbone, termed *representations*, and the projection head, known as *embeddings*, we train several models with various configurations of projection heads against SSL objectives on the ImageNet-100 dataset and calculate incoherences of representations and embeddings along with respective downstream accuracies. The performance of the representations and embeddings of the pre-trained model is evaluated in a downstream classification task, while incoherence $\mu$ for both candidates is estimated on the training set.

Since the dimensionality of representations and embeddings differ, i.e. 512 for backbone output and 2048 for projection head output, we need to pick the rank to compute $\mu$ accordingly. The common way to pick rank in the numerical methods is through tolerance, i.e. thresholding based on the values of the normalized singular values, i.e. $\sigma_i' = \sigma_i / \sum_j \sigma_j$. Otherwise, one could use the notion of effective rank [30], given by $r_e(\mathbf{A}) = \exp(H(\boldsymbol{\sigma}))$, where the entropy $H(\boldsymbol{\sigma}) = -\sum_i \boldsymbol{\sigma}_i \log(\boldsymbol{\sigma}_i)$ with vector of singular values $\boldsymbol{\sigma}$. Overall, we compute coherence as the following expression:

$$\mu(\mathbf{A}) = \frac{n}{r_e(\mathbf{A})} \max_{1 \leq i \leq \lceil r_e \rceil} \|\mathbf{U}_{\mathbf{A}}^\top \mathbf{e}_i\|_2^2.$$

**Incoherence and projection head complexity.** To estimate coherence of representations in Figure 2 (left), we embed the training set of ImageNet-100 to get representations matrix $\mathbf{A} \in \mathbb{R}^{125952 \times 512}$ and compute incoherence $\mu(\mathbf{A})$ using effective rank $r_e(\mathbf{A})$. We embed the training set using three distinct pre-trained models for each of the projection head configurations characterized by the number of layers $l$, $l \in \{1, 2, 3, 4\}$, and average the values across ten embedding runs. The resulting mean and standard deviation plot suggests that incoherence (low $\mu$) is higher for more shallow projection heads and decreases ($\mu$ grows) as the number of layers in the head increase, a result we anticipated.

**Embeddings and representations disparity.** Figure 2 (right) plots distinct models representations and embeddings in the Accuracy-Coherence plane. It encodes different type of loss with a shape: diamonds for VICReg and squares for RQmin. While blue colour signifies the position of each model's representations, the green colour reflects the corresponding embeddings. Both objectives demonstrate a *separation* of the embedding (low coherence, low accuracy) and representation (high coherence, high accuracy) points, which is more distinctive in the case of RQmin (squares).

The resulting plots support our hypothesis in Proposition 4.3 that incoherence indeed plays a crucial role in explaining the use of the backbone outputs. For successful matrix completion, high incoherence of the partially observed affinity matrix is essential. However, for the downstream performance of the representations, the opposite is preferred. The projection head functions as a disentangling buffer, enabling the representations to maintain low incoherence. Conversely, the embeddings inherit the incoherence of the affinity matrix.

While incoherence seems to be an attractive candidate for unsupervised embedding evaluation metric in scenarios with little to no test data, one should use caution as the coherence value $\mu$ does not reflect

the amount of relevant information stored in the matrix of embeddings due to normalization with reference to its rank. This idea has been explored in [35] on embeddings of supervised models.

# 6   Related Work

Recent success of self-supervised methods [9, 15, 2, 38, 39], especially in the domain of computer vision received great amount of attention since the learned representations ended up respecting the semantics of the data. There has been a great interest in trying to understand the inner workings of the seemingly heuristic objectives since.

While there are many lenses one may take up to study the problem [21, 37, 22, 34, 33, 18], a particularly related to this work concurrent body of literature has adopted the view of the kernel or laplacian-based spectral representation learning [13, 1], which we also share in this work. We highlight our main difference to the results provided in these very recent papers. [1] does a brilliant job connecting and characterizing modern SSL methods into classical existing counterparts. However, it does not answer an important question whether an incomplete a priori knowledge about the data manifold stemming from augmentations can provide a good approximation to essentially nonlinear dimensionality reduction methods such as LLE [29], MDS [24], and kernel PCA [31].

We not only show SSL methods to have an objective function similar to the objectives of classical spectral decomposition methods, e.g. LaplacianEigenmaps [3], but also try to address the problem of incomplete and noisy measurements that we get as an inductive bias during the augmentation process. We hope that this perspective via the low-matrix completion problem will yield further theoretical results on the success of self-supervised learning and practical benefits when applying these methods in the wild, e.g. in domains such as medical imaging [16] and hard sciences [25].

# 7   Conclusion and Future Work

In this work, we make an attempt to bridge modern self-supervised methods with classical Laplacian-based dimensionality reduction methods and low-rank matrix completion in hopes to provide theoretical insights on the recent successes of SSL methods.

We show that these methods are not only doing Laplacian-based nonlinear reduction but are able to approximate and recover the truncated version of the underlying Laplace operator given only noisy and incomplete information from augmentation protocol by adopting low-rank matrix completion extensive literature and results. However, when working with datasets with potentially large number of classes, it might be a good idea to consider whether the size of the sample is large enough so that the minimal cluster size allows the full data matrix to be considered low-rank, otherwise the SSL methods would possibly fail to converge.

We also spot a direct influence of the inductive bias in the parameterization of the learned map on the column space of the recovered matrix. We also hypothesize that the disparity in downstream performance between backbone and projection head outputs can be explained by the high incoherence if the latter which is tied to the incoherence of the kernel one is recovering during training. The kernel should have high incoherence to be recoverable.

One of the possible avenues for future work stems from the notion of incoherence. We see it in exploring incoherence property of different types of the similarity or weighting functions one may use to instantiate the adjacency matrix with.

We hope that this work paves the way for a deeper study of the connection between self-supervised methods and classical problems such as matrix completion to yield better practical and theoretical understanding of various applications of SSL in different domains, not only computer vision.

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
