# OpenReview forum: "Neural Harmonics: Bridging Spectral Embedding and Matrix Completion in Self-Supervised Learning"
_NeurIPS.cc/2023/Conference — NeurIPS 2023 poster_

### Official Review · Reviewer_2Q8M · 2023-07-03

**Soundness:** 3 good
**Presentation:** 3 good
**Contribution:** 2 fair
**Rating:** 5
**Confidence:** 3

**Summary:**

In this paper, the authors provide theoretical analysis on convergence and downstream performance of self supervised representation learning (SSL) approaches using tools from low-rank matrix completion.

In particular, (i) they relate an eigenproblem objective to SSL methods, (ii) find that SSL methods perform a conjunction of laplacian embedding and low rank matrix completion, (iii) relate SSL augmentations to the number of observed entries required in matrix completion and (iv) provide some experimentation around incoherence as it relates to downstream performance.

**Strengths:**

- Writing and theoretical exposition is clear
- Conceptually the trace maximization formulation is elegant. Specifically, the authors provide a broad framework through this line of reasoning to relate commonly used SSL objectives.
- The results on incoherence are well-aligned to empirical findings on projection head vs backbone representations, provides a potential explanation for this phenomena

**Weaknesses:**

- Additional experimentation on performance across the trace maximization approach for small scale datasets would be helpful
- Further experimentation for the incoherence results in Figure 2 would be helpful, for instance how does incoherence compare across different SSL methods  (i.e. SimCLR, BarlowTwins, etc)

**Questions:**

- Details on the experimentation run for CIFAR-10 would be helpful. Only a few short paragraphs are presented which claim equivalent performance to VICReg
- Are there any ways to make use of distributed low-rank matrix completion approaches with theoretical guarantees? (i.e Mackey, JMLR 2015)

**Limitations:**

- It is unclear how the proposed trace maximization scales in runtime with $N$ and the number of view augmentations
- Some of the assumptions only hold for the self-supervised case and not the supervised contrastive learning case (i.e. CLIP) such as positive anchor backbone representations being close to one another.

---

> ### Author Rebuttal · Authors · 2023-08-10
>
> We thank the reviewer for their insightful feedback!
>
> We would like to start the response by addressing the weaknesses pointed out by the reviewer. We first report additional experiments and details on the performance across small-scale datasets (CIFAR-10, CIFAR-100, and ImageNet-100), followed by additional experimentation for incoherence results. All results are summarized in the attached PDF for convenience. Hopefully, the following answers the concerns raised in the questions and limitations sections of the review. We finish by discussing the distributed low-rank matrix completion approach.
>
> We have conducted additional experiments for the trace formulation on CIFAR-10, CIFAR-100, and ImageNet-100 along with a comparison to VICReg, one of the commonly used SSL methods. The setup is equivalent to one given in the supplementary materials, we reiterate it here. We use the ResNet-18 backbone with a 3-layer projector with the following dimensions: 2048-2048-2048. Number of the training epochs is 1000 for CIFAR and 400 for ImageNet-100. The batch size for CIFAR-10 and CIFAR-100 is 256, and for ImageNet-100 — 512. We use the LARS optimizer: learning rate = 0.3, weight decay = 1e-4, and momentum = 0.9. The learning rate schedule is cosine annealing with a linear 10-epoch warmup. These hyperparameters are typical for VICReg training (thus favouring VICReg) and we do not adjust them to train our trace maximization formulation.
>
> While VICReg has 3 special hyperparameters that weigh individual loss terms, our trace formulation only requires specifying parameter $t$ of the heat kernel, which controls the mass spread across views in a positive pair/set. We fix $t=2$ across all experiments.
>
> We train each model on a single Nvidia V100 GPU with 16Gb memory. On CIFAR-100, VICReg runtime is 6h 7m, trace formulation - 5h 57m (similar timing difference for other datasets). The trace formulation does not incur additional runtime overhead. Since the approximate heat kernel matrix used in the trace maximization objective is sparse (essentially comprised of $m+1$ unique entries, where $m$ is the number of augmentations used) and can be stored and multiplied efficiently, the method scales similarly to other SimCLR-like methods with increasing batch size and number of augmentations. The major bottleneck for such methods is usually data preprocessing as augmentations are typically CPU-intensive.
>
> Below, we report the linear evaluation results on the trained models — mean accuracy across 5 and 4 independently learned models (top-1 / top-5) for CIFAR-10 and CIFAR-100, respectively, and 1 run for ImageNet-100 (we hope to add more training runs as they should be finished before the final discussion period ends). The evaluation protocol is standard (please see details in Appendix 8.2).
>
> |      | CIFAR-10 | CIFAR-100      | ImageNet-100 |
> | ----------- | ----------- | ----------- | ----------- |
> | VICReg      | 91.15 / 99.64       | 66.76 / 89.39      | 79.28 / 94.64       |
> | Ours   | 91.19 / 99.67        | 67.35 / 89.91   | 78.36 / 94.3        |
>
> These results are comparable to state-of-the-art even without any tuning for more favourable hyperparameters.
>
> Please see the attached PDF for Figure 1 and its caption reporting results on the incoherence across three commonly used SSL methods: SimCLR, BarlowTwins, and VICReg.
>
> Regarding the question about distributed low-rank matrix completion, indeed, there is an immediate connection between distributed low-rank MC and the stochastic gradient descent in SSL in the sense that they both perform a random sampling of the submatrices. While the distributed MC performs optimal approximations for all of the sampled submatrices simultaneously and adopts their averaging as a final step, in SSL typically one processes random batches sequentially. While the goal of the latter method is to provide matrix reconstruction or its low-rank factors, SSL aims to learn the parameterized function that computes a set of eigenfunctions at a given point. We could adopt the distributed approach in SSL since the underlying approximation step is differentiable as we still need to facilitate a learnable mapping.
>
> This review was particularly helpful in enhancing the submission and we hope that our response comprehensively addresses the concerns the reviewer had regarding our work.

---

### Official Review · Reviewer_LpdS · 2023-07-04

**Soundness:** 4 excellent
**Presentation:** 3 good
**Contribution:** 4 excellent
**Rating:** 7
**Confidence:** 3

**Summary:**

Self-supervised learning methods can effectively leverage limited signals to converge towards meaningful representations, but how is it made possible? This paper tries to give a response. This paper establishes a connection between SSL and a matrix completion problem by showing that these are Lagrangian dual of each other. This further implies that optimizing the SSL objsective simultaneously entails reconstructing the kernel matrix. This leads to some theoretical findings, including:

- The trace maximization formulation entails several popular SSL methods: SimCLR, BarlowTwins, VICReg.
- A less incoherent matrix is easier for matrix recovery, which explains why typical SSL methods rely on the representations (the incoherence is low) rather than the embeddings (the incoherence is high).

Based on the theoretical insights, this paper proposes a trace maximization objective for SSL (eq. 4). Following are some findings from experiments:

- The trace maximization objective is on par with existing SSL methods.
- There is a negative correlation trend between the incoherence and the number of layers in the projection head.
- The experiment findings support the hypothesis in proposition 4.3 — that incoherence indeed plays an important role in explaining the use of the backbone outputs.

**Strengths:**

- This paper proposes a novel, matrix completion formulation for SSL that can entail several popular SSL methods.
- The analysis of the matrix completion formulation provides insights about various parts of SSL.
- Empirically, the trace maximization objective leads to performances on par with other approaches.

**Weaknesses:**

I have some minor points about the usage of terminologies — please refer to the comment section below.

**Questions:**

- Figure 2: What is RQ loss?
- I’m not familiar with the SSL literature (I’m working on NLP), so the roles of the representation vs embedding appears differently from what I thought. Apparently, the representation is directly acquired from the backbone layers and is closer to the input. The embedding is further away from the input. In NLP, however, the embedding is the one that is closer to the input. But once I get my head around this discrepancy of terminologies, I see that many intuition and findings of this paper makes sense.

**Limitations:**

I do not see negative potential societal impacts of this work.

---

> ### Author Rebuttal · Authors · 2023-08-10
>
> We thank the reviewer for their perceptive feedback.
>
> The review offers a very well-structured and observant summary of the submission. We would like to add an additional comment about the incoherence and the role of the projection head. We think there might be a typo in the following ---
> *''A less incoherent matrix is easier for matrix recovery, which explains why typical SSL methods rely on the representations (the incoherence is low) rather than the embeddings (the incoherence is high)''*,
> but to be on the safe side, we would like to elaborate more.
>
> Higher incoherence intuitively means that information is *not* stored in the few important entries. Randomly sampling an incoherent matrix for a few of its entries will reveal more information than sampling a coherent one, which renders low-rank incoherent matrices possible to complete. Since common SSL methods train representations such that they recover spectral decomposition, the embeddings inherit this presumably high incoherence. But our proposition is that higher incoherence of representations makes them hard for downstream tasks. We posit that a projection head might play a buffer role for representations to be less incoherent and thus easier for downstream tasks.
>
> Next, we answer the reviewer's questions:
> - The RQ loss is a naming artifact, it denotes the trace maximization formulation. We apologize for this typo! We will change the notation accordingly.
> - Indeed, the terminology in NLP and SSL diverges in this regard causing confusion. We tried to compensate for that by reiterating the meaning of the used terminology in the submission as often as possible. We want to thank the reviewer for their patience and persistence!

---

> > ### Comment · Reviewer_LpdS · 2023-08-17
> >
> > Thank you for the clarification and answers. I am keeping the original score.

---

### Official Review · Reviewer_JcEa · 2023-07-05

**Soundness:** 3 good
**Presentation:** 3 good
**Contribution:** 3 good
**Rating:** 6
**Confidence:** 4

**Summary:**

This paper aims to provide a theoretical understanding of the recent successes of self-supervised learning methods by leveraging tools like Laplacian-based dimensionality reduction methods and low-rank matrix completion. The authors introduce an eigen-problem objective for spectral embeddings from graphs, which is used to interpret modern self-supervised learning methods.

**Strengths:**

1. Using an eigenproblem objective for spectral embeddings derived from graph augmentations, the authors explain the workings of contemporary self-supervised learning methods. This offers a fresh lens to understand self-supervised representation learning.
2. The paper further shows that self-supervised learning techniques can concurrently execute Laplacian-based nonlinear dimensionality reduction and low-rank matrix completion. This dual functionality further explains the success of self-supervised learning methods.


**Weaknesses:**

The approach presented in the paper is significantly dependent on the incoherence between the outputs of the backbone and the projection head.

**Questions:**

Could this approach help to understand the inner working of Large Language models (LLMs)?

**Limitations:**

The paper needs more numerical experiments.

---

> ### Author Rebuttal · Authors · 2023-08-10
>
> We thank the reviewer for their thoughtful feedback!
>
> First, we would like to address the concern about the dependence on the incoherence between the backbone and the projection head.
>
> We first would like to clarify that the underlying assumption for learning useful representations is that the similarity/kernel matrix of the data is somehow aligned with the downstream task. For example, if the measure of similarity between objects is ‘orthogonal’ to the downstream task, then we cannot hope to extract useful information into the representations.
>
> However, there is another aspect of whether we can produce useful representations. In this work, we argue that modern SSL methods perform spectral embedding and low-rank matrix completion. Successful matrix completion relies on the high incoherence of the matrix we want to complete. As all the methods considered in the submission could be seen as performing spectral decomposition, the produced representations inherit the incoherence of that matrix. We argue that this incoherence phenomenon could explain why the projection head is not used in the downstream tasks.
> All in all, we could summarize the above by saying that the problem itself and thus all considered SSL methods assume and rely on the underlying similarity/kernel matrix to be incoherent and thus recoverable.
>
> We would also like to note that we conducted more experiments with our trace maximization formulation and present the results in the PDF attached to the global response.
>
> Regarding the question about the inner workings of the large language models, we are enthusiastic to explore if this approach could offer insights into the representations learned by LLMs — one may find many similarities in the sense many NLP approaches are based on pretext tasks. However, now we cannot meaningfully elaborate on this matter further.

---

> > ### Comment · Reviewer_JcEa · 2023-08-14
> > **Comments after rebuttal**
> >
> > Thank you for your response. I am keeping my overall recommendation at 6.

---

### Official Review · Reviewer_1mcz · 2023-07-05

**Soundness:** 4 excellent
**Presentation:** 3 good
**Contribution:** 4 excellent
**Rating:** 8
**Confidence:** 4

**Summary:**

The authors observe that self-supervised learning (SSL) attracts growing attention and that, by now, numerous corresponding loss functions have been proposed. They systemize these from the point of view of Laplace operators (on Riemannian manifolds) and low-rank matrix approximation. Indeed, for SimCLR, BarlowTwins, and VICReg they show that these learn eigenfunctions of a Laplacian. They also demonstrate that models trained w.r.t. related trace maximization objectives reach performances that are on par with those resulting from modern SSL techniques.

**Strengths:**

This paper is a blast from the past in the best possible sense. More rigorous theoretical underpinnings of recent self-supervised learning models are still very much lacking and this paper makes considerable progress in this regard and shows a connection to low-rank-matrix completion tasks. This contribution is theoretically rigorous and technically sound and solid. It also shows that “well known” trace maximization objectives lead to model which reach performances that are on par with those resulting from modern SSL techniques.

**Weaknesses:**

A minor point of criticism is that several interesting experimental findings are deferred to the supplementary material.

**Questions:**

-

**Limitations:**

Given the scope of the paper (bridging a gap between modern learning techniches and classically "well known" models), I can't see any practical limitations. Neither are there any concerns regarding negative societal impact.

---

> ### Author Rebuttal · Authors · 2023-08-10
>
> We thank the reviewer for their kind and perceptive feedback.
>
> We sincerely appreciate the reviewer’s high opinion of the strengths of this submission. We hope that additional experimentation results (attached as PDF, summarized in the global response) will only help reinforce this position. We will find a way to incorporate all interesting experimental findings in the main body.

---

> > ### Comment · Reviewer_1mcz · 2023-08-21
> >
> > Thanks for your response; I am keeping my score.

---

### Author Rebuttal · Authors · 2023-08-10

We would like to thank all the reviewers for their time, effort, and considerate reviews!

While we will address each review individually, in this global response, we would like to summarize the main pieces of those individual responses.

First of all, we would like to present more experimental results concerning the performance of the trace objective and the incoherence tested on common methods (SimCLR, BarlowTwins, and VICReg). We have compiled them into a PDF for your convenience. There, Table 1 reports the top-1 and top-5 downstream accuracy (mean and standard deviation across 3-5 runs) of our proposed formulation (Ours) and VICReg across three datasets: CIFAR-10, CIFAR-100, and ImageNet-100. In short, the performance of the proposed objective is comparable to the state-of-the-art.

Meanwhile, Figure 1 reports incoherence vs accuracy results for SimCLR, BarlowTwins, and VICReg on the same architecture and training/testing configuration. All methods reveal similar behaviour — backbone outputs are less incoherent and have better downstream performance, while projection head outputs have high incoherence and worse downstream performance. We caution, however, that incoherence should not be used to compare methods/models since it is not indicative of the information load in the representations/embeddings, which would typically be measured by metrics based on matrix rank.

While we do not necessarily strive to overtake the state-of-the-art — we do not optimize the hyperparameters in our favour, we would like to emphasize the importance of establishing the connection of augmentation-based self-supervised methods to spectral embedding methods and low-rank matrix completion, which our submission aims to provide. This connection also hints at the reason why one typically disposes of the projection head once the model is trained and used in a downstream task. We posit that incoherence of the matrix that one tries to recover is behind it and find empirical support for this phenomenon.

The reviews were instrumental in enhancing the presentation, shaping new experiments, and finding additional insights. We hope that our response provided better clarity on the submission, resolved any confusing aspects, and provided convincing reasons for a potential score increase.

---

### Decision · Program_Chairs · 2023-09-21

**Decision:**

Accept (poster)

**Comment:**

This paper studies SSL methods & their effectiveness by providing a theoretical analysis using Laplacian and low-rank matrix completion methods. The objective is well motivated, proposed approach is definitely a nice & stand-out contribution (among a plethora of existing empirical results from common “transformer-X applied to task-Y” style studies), and a good direction towards better understanding why SoTA SSL methods “just” work. The authors also do a nice job in clarifying & addressing some of the concerns raised by reviewers in their comments such as runtime complexity, the effect of incoherence across different settings.

A few areas & suggestions where the paper can improve and benefit from more detailed discussion/analysis is a) provide empirical results on more benchmark datasets/tasks, b) validate how the expected behavior conforms (or not) for larger models & more complex  tasks, c) adding more details (perhaps in future / discussion section) on the connection w/ approximations (distributed low-rank MC and other approx. from MC literature) & how this could be extended to other domains/tasks (for instance, when there is structure in the output space)

Overall, this is an interesting work & direction and the authors do a good job wrt coming up with a nice theoretical formulation to study the SSL efficacy. The authors are encouraged to use all the provided feedback and incorporate suggestions/comments from discussions to improve the final version of the paper.